# Cytotoxic Tph subset with low B-cell helper functions and its involvement in systemic lupus erythematosus
Noriyasu Seki [1,2], Hideto Tsujimoto[1,2], Shuhei Tanemura[1,2], Shinji Kojima[3], Fumihiko Miyoshi[3], Jun Kikuchi [2], Shuntaro Saito[2], Mitsuhiro Akiyama[2], Kunio Sugahara[1,2], Keiko Yoshimoto[2], Yuko Kaneko[2], Kenji Chiba [1,2] ✉ & Tsutomu Takeuchi [2,4]

T peripheral helper (Tph) cells are thought to contribute to extra-follicular B cell activation and play a pathogenic role in autoimmune diseases. However, the role of Tph subsets is not fully elucidated. Here, we investigate the immunological functions of Tph subsets and their involvement in systemic lupus erythematosus (SLE). We have defined four Tph subsets (Tph1: CXCR3+CCR6−, Tph2: CXCR3−CCR6−, Tph17: CXCR3−CCR6+, and Tph1-17: CXCR3+CCR6+) and performed RNA sequencing after cell sorting. Tph1 and Tph17 subsets express substantial levels of *IL21*, indicating B cell helper functions. However, Tph2 and Tph1-17 subsets express low *IL21*. Interestingly, we have found Tph2 subset express high levels of *CX3CR1*, *GZMB*, *PRF1*, *GLNY*, *S1PR5*, *TBX21*, *EOMES*, *ZNF863*, and *RUNX3*, indicating a feature of CD4+ cytotoxic T lymphocytes. In SLE patients, the frequency of Tph1 and Tph2 subsets are significantly increased and positively correlated with SLE disease activity indexes. Tph1 cells expansion has been observed in patients with cutaneous and musculoskeletal manifestations. On the other hand, Tph2 cell expansion has been found in patients with lupus nephritis in addition to the above manifestations. Our findings imply that Tph1 and Tph2 subsets exert distinct immunological functions and are contributed to the complexity of clinical manifestations in SLE.

Systemic autoimmune diseases such as rheumatoid arthritis (RA) and systemic lupus erythematosus (SLE) are characterized by T cell activation driving the production of pathogenic class-switched autoantibodies[1]. Activated CD4+ T cells in SLE are correlated with autoantibody production and disease activity, suggesting their role in promoting disease[1,2]. An understanding of pathways driving heightened T cell activation in SLE are necessary to identify and define therapeutic targets. SLE is characterized by multiple phenotypes and multi-tissue damage, and typical clinical manifestations associated with SLE are arthritis, fever, photosensitivity, malar rash, lymphopenia, Raynaud phenomenon, serositis, nephritis, and neurological involvement[3–5]. The production of heterogeneous autoantibodies is thought to involve in the pathogenesis of SLE[3,4]. Particularly, lupus nephritis is one of the critical manifestations with impact to mortality[6].

A CD4+ T cell subset, peripheral helper T (Tph) cells expressed high level of programmed cell death-1 (PD-1), inducible T cell co-stimulator (ICOS), and HLA-DR and can induce plasma cell differentiation through interleukin (IL)-21 secretion[7]. Notably, Tph cells lack C-X-C motif chemokine receptor (CXCR) 5, a defining marker for T follicular helper (Tfh) cells, and instead express C-C motif chemokine receptor (CCR) 2, CCR5, and C-X3-C motif chemokine receptor (CX3CR)1 that direct migration to inflamed sites[7,8]. Tph cells were found to be expanded in synovium from patients with RA[7,9] and associated with the pathogenesis and disease severity[10–12]. In patients with SLE, Tph cells were shown to be increased significantly in the blood and were positively correlated with the disease activities[13–15]. Our previous studies showed that type I and III interferons (IFNs) promote differentiation of IL-21-producing-Tph-like cells from naïve CD4+ cells[16,17]. We also found that the expansion of Tph cells was

[1]Research Unit Immunology & Inflammation, Innovative Research Division, Mitsubishi Tanabe Pharma Corporation, Yokohama-shi, Kanagawa, Japan. [2]Division of Rheumatology, Department of Internal Medicine, School of Medicine, Keio University, Shinjuku-ku, Tokyo, Japan. [3]Discovery Technology Laboratories, Innovative Research Division, Mitsubishi Tanabe Pharma Corporation, Yokohama-shi, Kanagawa, Japan. [4]Saitama Medical University, Iruma-gun, Saitama, Japan. ✉e-mail: Chiba.Kenji@mk.mt-pharma.co.jp

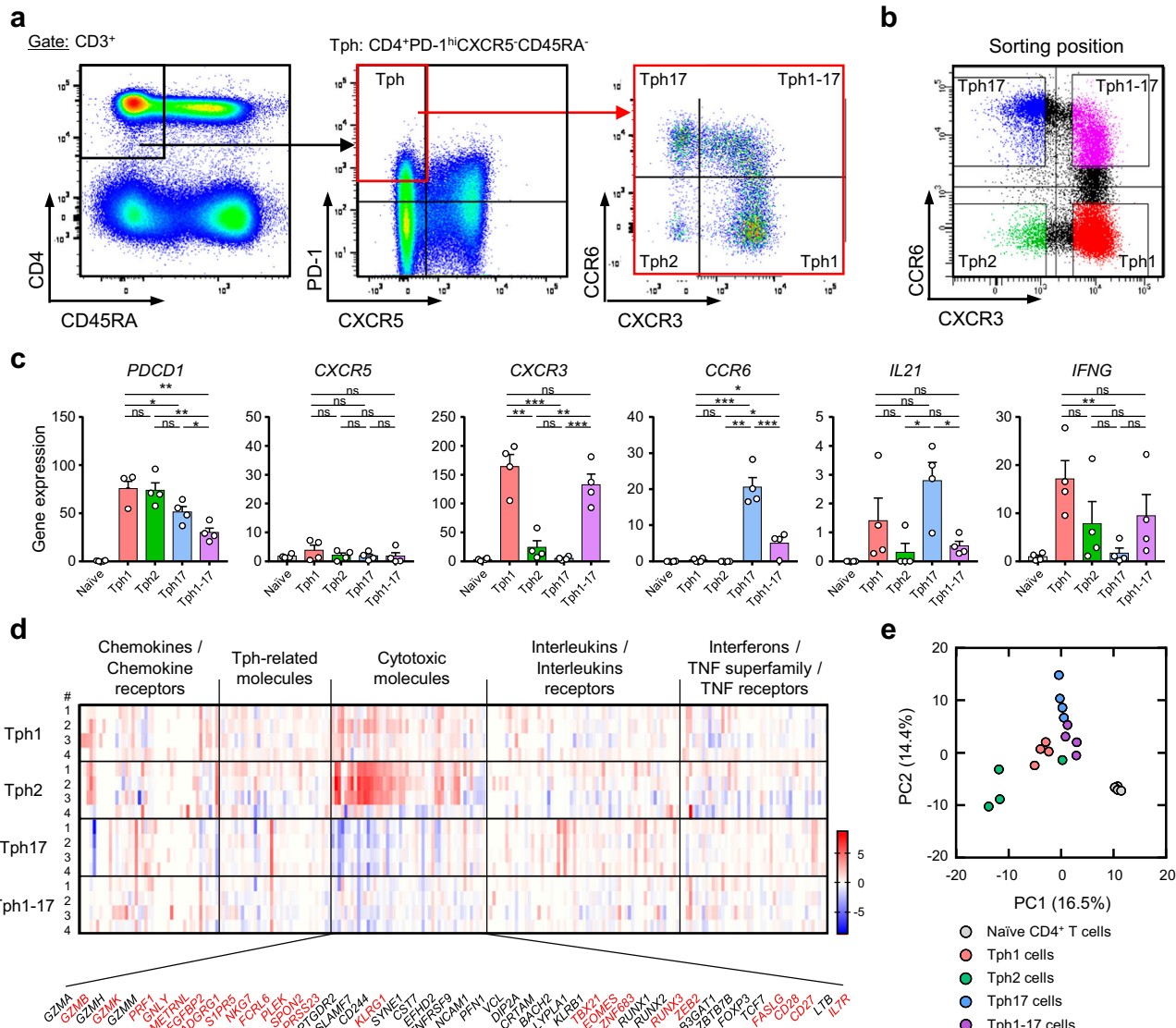

**Fig. 1 | Tph2 subset shows a unique gene expression pattern as compared with Tph1, Tph17, and Tph1-17 cells. a** Gating strategy of CD45RA⁻ memory CD4⁺ T cells, PD-1^hiCXCR5⁻ Tph cells, and Tph subsets (Tph1, CXCR3⁺CCR6⁻; Tph2, CXCR3⁻CCR6⁻; Tph17, CXCR3⁻CCR6⁺; Tph1-17, CXCR3⁺CCR6⁺). **b** Separation of four Tph subsets by cell sorting. **c** The expression of *PDCD1, CXCR5, CXCR3, CCR6, IL21,* and *IFNG* in naïve CD4⁺ T cells and Tph subsets from healthy donors

$(n = 4)$. Data represent the mean ± SEM of 4 independent donors. Data were statistically analyzed using unpaired *t*-test. *$P < 0.05$, **$P < 0.01$, ***$P < 0.001$, ns: not significant. **d** The expression patterns of selected 357 genes (Supplementary Table 1) of immunoregulatory molecules in each Tph subset $(n = 4)$ by RNA-sequencing after cell sorting. **e** PCA of each Tph subset $(n = 4)$ based on the selected 357 genes.

positively correlated with serum IFN-α or IFN-λ1 in SLE, suggesting that type I and III IFNs trigger expansion of Tph cells[17]. Tph cells are shown to be associated with the pathogenesis of primary Sjögren's syndrome (p-SS)[18,19], anti-neutrophil cytoplasmic antibodies (ANCA)-associated vasculitis (AAV)[20], dermatomyositis[21], and IgG4-related disease (IgG4-RD)[22,23]. These results imply that Tph cells have a B cell helper function and promote extrafollicular autoantibody production and play a pathogenic role in various autoimmune diseases.

It has been reported that Tph subsets are classified by the cell surface expression patterns of CXCR3 and CCR6 and are involved in the disease activities and autoantibody productions in SLE[14,15,24]. However, it remains unclear whether Tph subsets have similar immunological functions and which Tph subsets are involved in various clinical manifestations of SLE. In this study, to understand the immunological functions of Tph subsets, we separated into each Tph subset by cell sorting and analyzed their gene expressions by RNA sequencing. In addition, we investigated the frequency of Tph subsets within memory CD4⁺ T cells in the blood and analyzed the association between Tph subsets and various clinical manifestations in SLE patients.

## Results

### Tph2 subset shows a unique gene expression pattern as compared with Tph1, Tph17, and Tph1-17 cells

We determined PD-1^hiCXCR5⁻ Tph cells within CD45RA⁻ memory CD4⁺ T cells and subsequently defined 4 Tph subsets (Tph1, Tph2, Tph17, and Tph1-17) based on the cell surface expressions of CXCR3 and CCR6 (Fig. 1a). To elucidate the immunological functions of Tph subsets, we separated 4 Tph subsets (Fig. 1b) and naïve CD4⁺ T cells from peripheral blood lymphocytes of healthy donors by cell sorting and by isolation kit, respectively, and performed the RNA sequencing of these T cells. We confirmed the high expression of *PDCD1* and low expression of *CXCR5* in all four Tph subsets by comparing with naïve CD4⁺ T cells (Fig. 1c, Supplementary Table 1). Tph1 and Tph17 subsets expressed relatively high levels of *CXCR3* whereas Tph17 and Tph1-17 subsets did relatively high *CCR6* when compared to the other Tph subsets, indicating a successful separation of Tph subsets (Fig. 1c, Supplementary Table 1). Tph1 and Tph17 subsets expressed substantial levels of *IL21* as compared with naïve CD4⁺ T cells; however, Tph2 and Tph1-17 subsets expressed only low level of *IL21* (Fig. 1c).

Noticeably, Tph2 subset in 3 out of 4 individuals showed no detectable *IL21*. Furthermore, Tph1 and Tph1-17 cells expressed the high levels of *IFNG*. We selected 357 genes of immunoregulatory molecules including cytokines, chemokines, their receptors, Tph-related molecules, transcriptional factors, and cytotoxic molecules (Supplementary Table 2). The expression patterns of the selected genes in Tph1, Tph2, Tph17, and Tph1-17 subsets are shown in Fig. 1d. Tph2 subset shows a unique gene expression pattern as compared with Tph1, Tph17, and Tph1-17 cells, and a most striking feature of Tph2 subset is increased gene expressions of cytotoxicity-related molecules including *GZMB, GZMH, PRF1, GLNY, TBX21, METRNL* (IL-41)*, FGFBP2, ADGRG1* (G-protein-coupled receptor GPR56)*, S1PR5, NKG7, FCRL6, PLEK, SPON2, PRSS23, KLRG1, SYNE1, TBX21, EOMES, ZNF683* (Hobit)*, RUNX3, ZEB2, and FASLG*. Among the selected genes of immunoregulatory molecules, the expression values greater than 0 in more than two samples were analyzed by principal component analysis (PCA). As shown in Fig. 1e, all points of naïve CD4$^+$ T cells were in almost the same regions. All points of Tph1, Tph17, and Tph1-17 cells were located to neighbor area. Interestingly, three points out of four Tph2 were located within the regions distinct from other Tph subsets and the remained one point of Tph2 was located near to the area of Tph1 and Tph1-17.

### Tph2 subset expresses high levels of cytotoxicity-related transcripts and low B cell helper functions

The pivotal gene expressions are shown in Fig. 2. The striking feature of Tph2 subset was high levels of *CX3CR1* (Fig. 2a). The expressions of *MAF* and *PRDM1* were relatively high levels in Tph1 and Tph17 subsets but the expressions of *BCL6* in all 4 Tph subsets were low levels and similar with naïve CD4$^+$ T cells (Fig. 2a). Noticeably, the levels of *GZMB, PRF1, GNLY, METRNL, FGFBP2, ADGRG1, S1PR5, NLG7, FCRL6, PREK, SPON2, PRESS23, and KLGR1* were relatively higher in Tph2 subset (Fig. 2b). Further, Tph2 subsets exhibited high expressions of *TBX21, EOMES, ZNF683, RUNX3, ZEB2, and FASLG*. On the other hand, Tph2 subset showed low expression of *CD27, CD28, IL7R, LAG3, HAVCR2* (TIM-3), and *CTLA4*, and almost comparable levels of *TIGIT* and *GATA3* with other Tph subsets (Fig. 2b, Supplementary Fig. 1), Unlike *GZMB* and *GZMH, GZMK* was markedly high in Tph1 subset.

Figure 3a shows heatmap of the gene expressions of pivotal cytokines, chemokine receptors, cytotoxicity-related molecules, and Tph-related molecules in each Tph subset. Totally, Tph1, Tph17, and Tph1-17 subsets showed a character of typical Tph cells with B cell helper functions. On the other hand, Tph2 subset expressed high levels of various cytotoxicity-related molecules but had relatively low expression of the molecules related to B cell helper function. These results strongly suggest that Tph2 subset shows cytotoxic functions and low B cell helper activity. Thus, we found that Tph2 subset shows a quite unique pattern of gene expressions suggesting a feature of CD4$^+$ cytotoxic T lymphocytes (CTL). We next investigated the association of transcript levels of molecules related to B cell helper functions, cytotoxic functions and transcriptional factors. The results of multiple correlation analyses are shown in Fig. 3b. The expression levels of *IL21* and *TNFSF13B* were positively correlated with those of *MAF* and *PRDM1* whereas showed negative correlations with those of *GZMH, PRF1, GLNY, TBX21*, and *EOMES* (Fig. 3b). More interestingly, the expression levels of *TBX21, EOMES, RUNX3, ZNF683, ZEB2, BCL6*, and *BATF* exhibited a positive correlation among themselves (Fig. 3c). From these results, it is highly probable that B cell helper functions and cytotoxic functions in Tph subsets are regulated by distinct transcriptional factors and that cytotoxic functions are predominantly regulated by *TBX21, EOMES, RUNX3*, and *ZNF683*.

### The expansion of Tph1 and Tph2 subsets in the blood of untreated SLE patients

Next, we determined the frequency of 4 Tph subsets (Tph1, Tph2, Tph17, and Tph1-17 cells) within CD45RA$^-$ memory CD4$^+$ T cells in the blood of patients with various autoimmune diseases including SLE and healthy controls (HC) (Supplementary Tables 3, 4). The percentages of Tph1, Tph2,

and Tph1-17 cells within memory CD4$^+$ T cells were markedly increased in the blood of SLE patients compared to HC (Fig. 4a, Supplementary Table 5). Notably, the percentages of Tph cells in untreated (new-onset) SLE patients were significantly increased as compared with untreated patients with RA, p-SS, AAV, or IgG4-RD. Further, the percentages of Tph2 cells were markedly increased in untreated patients with SLE, AAV or IgG4-RD compared to untreated patients with RA or p-SS. There was no clear change in frequency of Tph17 cells. The largest population of Tph subsets was Tph1 cells, the second one was Tph1-17 cells, and the smallest subset was Tph2 cells in HC whereas Tph17 cells in SLE (Fig. 4b). These results indicate that the increased frequency of Tph1 and Tph2 subsets are the feature of Tph cell expansion in SLE.

### Tph1 and Tph2 subsets play a pathogenic role in SLE

We investigated the correlation between the frequency of Tph subsets and several clinical parameters in SLE patients. The percentages of whole Tph cells as well as Tph1 and Tph2 subsets were positively correlated with SLEDAI, anti-double strand (ds)-DNA antibody (Ab) titers, proteinuria scores, the levels of complement 1q (C1q) (Fig. 5), and the neutrophil/lymphocyte ratio (Supplementary Table 6). Additionally, the percentages of whole Tph, Tph1, and Tph2 cells were negatively correlated with the levels of C3, C4, and 50% hemolytic unit of complement (CH$_{50}$). Alternatively, there was no clear correlation between Tph17 cells and clinical parameters including SLEDAI. The percentages of Tph1-17 cells were positively correlated with SLEDAI and anti-ds-DNA Ab titers and negatively correlated with C3, C4, and CH$_{50}$. These results suggest that Tph1 and Tph2 cells rather than Tph1-17 or Tph17 cells play a pathogenic role in SLE.

### Involvement of Tph1 and Tph2 subsets in clinical manifestations of SLE

To understand the involvement of Tph subsets in clinical manifestations of SLE, we compared the frequency of Tph subsets in patients with a clinical manifestation and ones without it. The percentages of Tph1 and Tph2 cells were significantly higher in SLE patients with cutaneous manifestation, musculoskeletal manifestation, serositis, or fever when compared with patients without each manifestation (Fig. 6a, Supplementary Table 7). Furthermore, SLE patients with nephritis showed significantly higher percentages of Tph2 cells when compared to patients without nephritis. On the other hand, high percentages of Tph17 or Tph1-17 subsets were limited to be seen in patients with only cutaneous or musculoskeletal manifestation.

When SLE patients with skin or musculoskeletal manifestations became lupus low disease activity state (LLDAS) by the treatment with medications, the frequency of Tph1 and Tph2 cells, like whole Tph cells, were reduced significantly as compared with that before the treatment (Fig. 6b, Supplementary Table 8). Furthermore, the percentages of Tph2 cells were also reduced markedly when patients with lupus nephritis became LLDAS (Fig. 6b). By contrast, the percentages of Tph17 cells were increased unexpectedly and those of Tph1-17 cells were only significantly reduced in patients with musculoskeletal manifestation. Additionally, the alterations of Tph subsets were not clear in case of patients who showed inadequate response for the medications (Supplementary Fig. 2, Supplementary Table 9). These results suggest that Tph1 and Tph2 subsets are mainly associated with cutaneous and musculoskeletal manifestations and Tph2 cells are additionally related to lupus nephritis also. Consequently, it is highly probable that Tph1 and Tph2 subsets shows different involvement in clinical manifestations in SLE.

### Different involvement of Tph1 and Tph2 subsets in clinical manifestations depends on their chemokine receptors

It has been reported that Tph cells, unlike Tfh cells, lack CXCR5 and instead express CCR2, CCR5, and CX3CR1 that direct migration to inflamed sites[7,8]. Based on these results, we analyzed the cell surface expression of CCR2, CCR5, and CX3CR1 on each Tph subset and determined the blood levels of several ligand chemokines. Typical staining patterns of CCR2, CCR5, and CX3CR1 on each Tph subset in HC and new-onset SLE patients are shown

in Fig. 7a. In one case of HC, Tph1, Tph17, and Tph1-17 subsets expressed significant levels of CCR5 and CCR2 but extremely low levels of CX3CR1, and Tph2 subset did low levels of these three chemokine receptors. On the

other hand, in a new-onset SLE patient, the most striking feature was a markedly increased expression of CX3CR1 on Tph2 cells (Fig. 7a). The cumulative results of the chemokine receptor expression patterns of Tph1

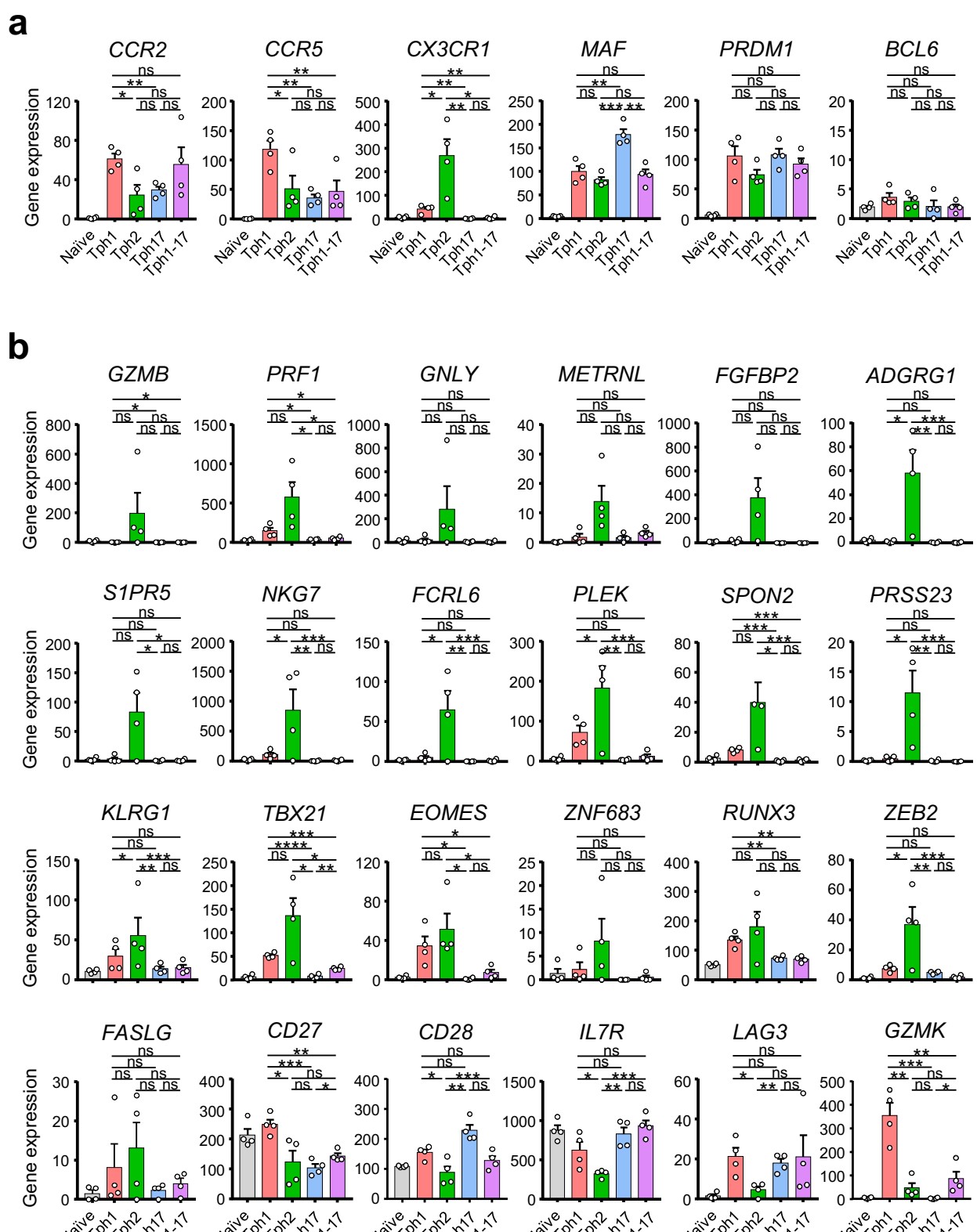

**Fig. 2 | Tph2 subset expresses high levels of cytotoxicity-related transcripts and low B cell helper activity. a** The pivotal gene expression of *CCR2, CCR5, CX3CR1, MAF, PRDM1,* and *BCL6* in each Tph subset. **b** The pivotal gene expression of cytotoxicity-related molecules and transcriptional factors in each Tph subset. Data

represent the mean ± SEM of 4 independent donors. Data were statistically analyzed using unpaired *t*-test. *$P < 0.05$, **$P < 0.01$, ***$P < 0.001$, ****$P < 0.0001$, ns: not significant.

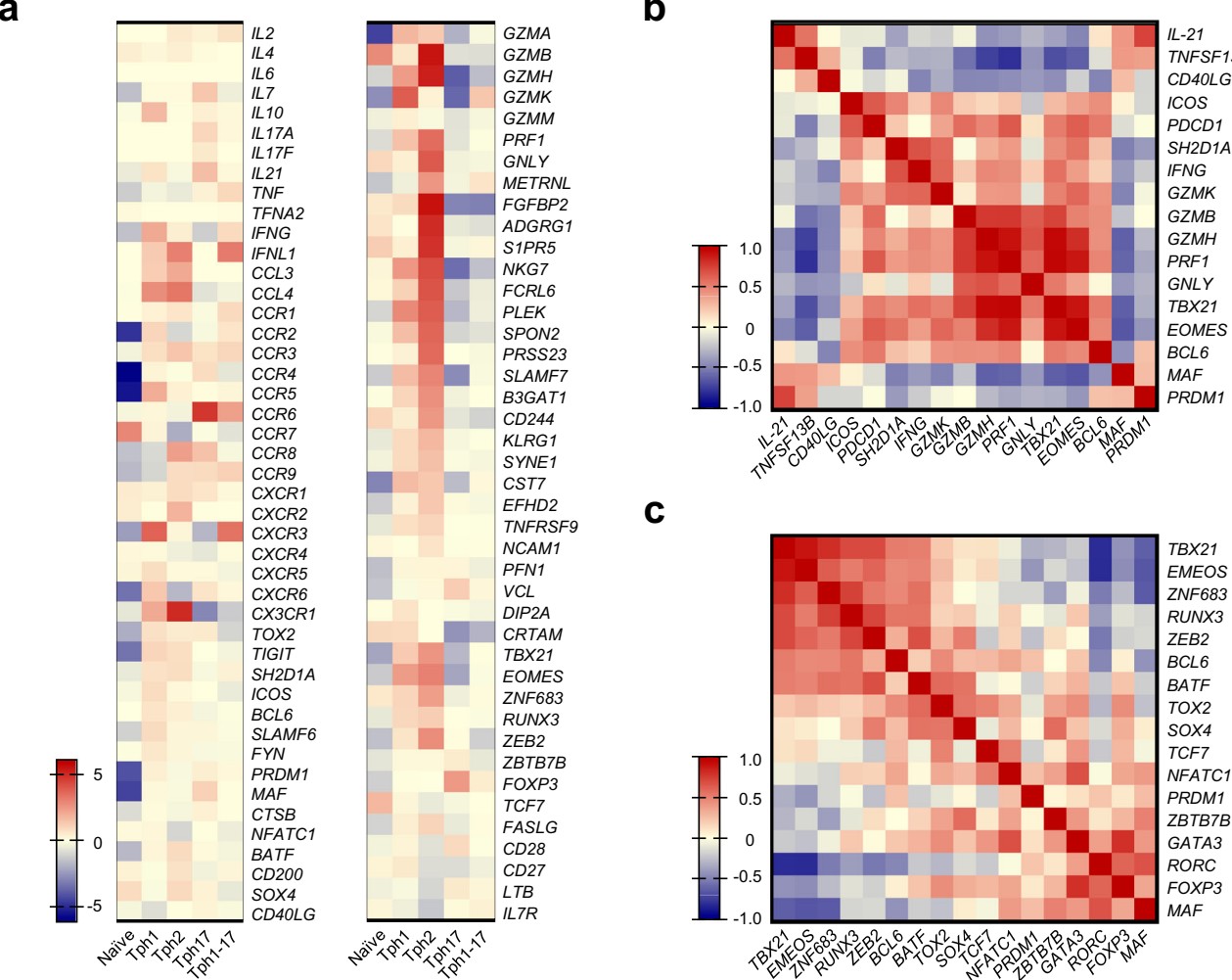

**Fig. 3 | The gene expression pattern of Tph2 subset shows a feature of CD4⁺ CTL. a** Heatmap of the pivotal gene expressions in each Tph subset. Results are shown as the mean fold change (log2) of median reduced gene expression levels. **b** Heatmap of multiple correlation among pivotal Tph-related and cytotoxicity-related molecules. **c** Heatmap of multiple correlation among pivotal transcriptional factors. The *r* values in Spearman's correlation coefficient are shown (**b**, **c**).

and Tph2 subsets in HC (*n* = 5) and new-onset SLE patients (*n* = 6) were shown in Fig. 7b. In HC, approximately more than half of Tph1 cells expressed CCR2 (55.1%) or CCR5 (59.4%) but CX3CR1-positive cells were low levels (12.8%) while Tph2 cells in HC had low expressions of these chemokine receptors (CCR2, 17.1%; CCR5, 22.5%; CX3CR1, 19.7%). In new-onset SLE patients, CCR5-positive Tph1 and Tph2 cells were increased by 59.0% and 51.8%, respectively. Notably, CX3CR1-positive Tph2 cells were markedly increased by 54.1% in new-onset SLE patients.

We further determined the levels of several chemokines in SLE patients and HC. The blood levels of CCL2, CCL3, CX3CL1, and CXCL10 were significantly higher in SLE patients compared to HC (Fig. 7c). In addition, there was no clear change in the levels of CCL4, CCL5, CCL13, CCL20, and CCL26 between HC and SLE, but those of CXCL9 and CXCL11 were significantly higher in SLE patients compared to HC (Supplementary Fig. 3). The frequency of Tph1 cells were positively correlated with the levels of CCL2, CCL3, and CXCL10 but not CX3CL1 (Fig. 7d). On the other hand, the frequency of Tph2 cells were positively correlated with the levels of CCL2, CCL3, and CX3CL1 (Fig. 7d). These results imply that Tph1 cells are mainly regulated by CXCR3⁻CXCL9/10/11 axis, and partially by CCR2-CCL2 axis and CCR5-CCL3 axis, while Tph2 cells are predominantly regulated by CX3CR1-CX3CL1 axis in SLE. Consequently, it is presumed that the different involvement of Tph1 and Tph2 subsets in clinical manifestations of SLE depends on the expression patterns of CCR2, CCR5, and CX3CR1.

### CX3CR1-positive Tph2 cells express granzyme B and perforin

Because the RNA sequencing analysis revealed that Tph2 cells express relatively high levels of *CX3CR1, GZB, PRF1, GNLY*, it is suggested that Tph2 cells has a character of CD4⁺ CTL. We next investigated the intracellular expression of granzyme B and perforin by flow cytometry. The Fig. 8a clearly shows that the only Tph2 subsets expressed both granzyme B and perforin within the cytoplasm. More than half of Tph1 cells (53.9 to 80.8%) were granzyme B- and perforin-double negative (Fig. 8b). By contrast, the majority of Tph2 cells (44.8 to 72.1%) expressed significant levels of both granzyme B and perforin (Fig. 8b). Notably, more than 75% of CX3CR1-positive Tph2 cells were granzyme B- and perforin-double positive cells (Fig. 8c). Thus, our results clearly demonstrated that CX3CR1-positive Tph2 cells has cytotoxic functions and namely has a feature of CD4⁺ CTL.

### Discussion

It has been well known that Tph cells can induce plasma cell differentiation through IL-21 secretion and concomitantly express cytotoxicity-related molecules such as *GZMB* and *PRF1*[7,8]. In this study, by RNA sequencing analyses of Tph subsets, we found that Tph1 and Tph17 subsets express substantial levels of *IL21*, however Tph2 and Tph1-17 subsets expressed low levels of *IL21* implying low B cell helper functions. Noticeably, Tph2 subset expressed CX3CR1, granzyme B, and perforin and therefore possesses cytotoxic functions, indicating a feature of CD4⁺ CTL.

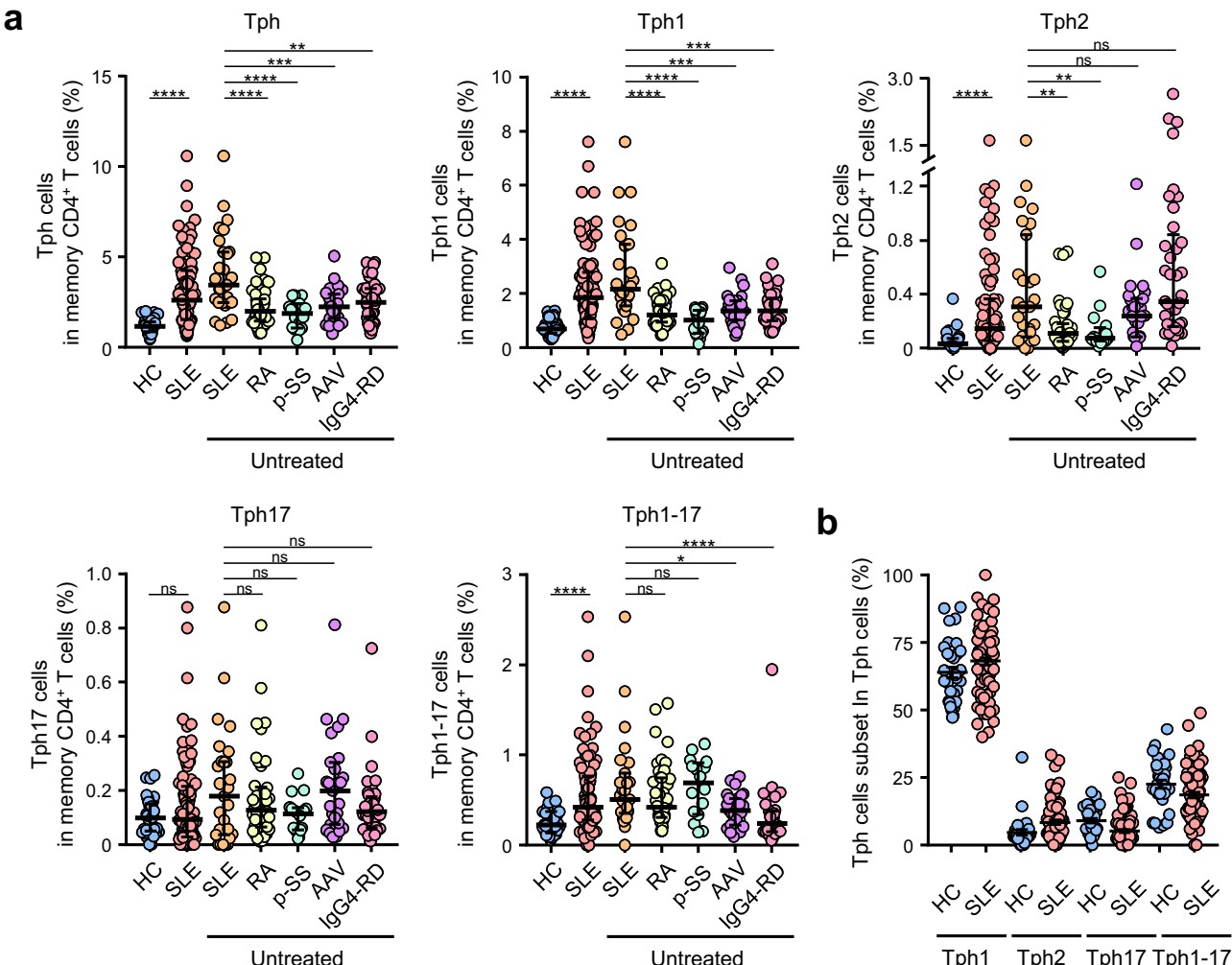

**Fig. 4 | The frequency of Tph subsets in blood memory CD4⁺ T cells of patients with autoimmune diseases including SLE. a** The proportions of each Tph subset in memory CD4⁺ T cells of HC ($n = 33$), SLE patients ($n = 85$), untreated patients with new-onset SLE ($n = 27$), RA ($n = 46$), p-SS ($n = 16$), AAV ($n = 27$), and IgG4-RD ($n = 37$). **b** The proportions of each Tph subset within whole Tph cells in SLE patients ($n = 85$) and HC ($n = 33$). Data represent the median ± IQR. Data were statistically analyzed using Mann-Whitney $U$-test. *$P < 0.05$, **$P < 0.01$, ***$P < 0.001$, ****$P < 0.0001$, ns: not significant (**a**, **b**).

Consistent with our findings, recent reported demonstrated that CX3CR1⁺ Tph-like cells with no capacity of IL-21 production abundantly possessed granzymes and perforin-1 in patients with IgG4-RD[22]. These results presumably indicate that distinct Tph subsets have different immunological functions.

More interestingly, Tph2 subsets exhibited relatively higher expressions of *TBX21*, *EOMES*, *RUNX3*, and *ZNF683* which are critical transcriptional factors for the differentiation of CD4⁺ CTLs[25,26]. Recent studies demonstrated that CD4⁺ CTLs represent a unique Th subset possessing antigen-specific cytotoxic activity in both humans and mice[27,28]. The strong association of the appearance of CD4⁺ CTLs with viral infections suggests an important role of this subset in antiviral immunity by controlling viral replication and infection[26]. Moreover, CD4⁺ CTLs are the lineage of CD45RA-positive CD4⁺ T effector memory cells (TEMRA)[27,28] and these cells might cause immunopathology in autoimmune diseases[25,26]. In contrast, Tph cells are basically classified as PD-1^hiCXCR5⁻ cells within CD45RA-negative memory CD4⁺ T cells and therefore it is highly likely that Tph2 cells with cytotoxic functions are distinguishable from CD4⁺ TEMRA.

Recently, it has been reported that GPR56⁺ Tph subset has been identified in the synovial fluid of RA patients with anti-citrullinated protein antibodies[29]. The frequency of GZMB⁺PRF1⁺GPR56⁺CD4⁺ T cells in synovial fluids positively correlated with cyclic citrullinated peptide positivity. Because our data revealed that Tph2 subset expressed higher levels of

*GZMB*, *PRF1*, and *ADGRG1* (GPR56) when compared to the other Tph subsets, it is highly likely that Tph2 subset are quite similar population as the reported GPR56⁺CD4⁺ T cells in RA. On the other hand, the synovial GPR56⁺CD4⁺ T cells in RA are shown to be expressed high levels of *LAG3*[29]. Furthermore, it has been shown that in chronic infections, cancer and autoimmune diseases, persistent antigen stimulation under suboptimal conditions can lead to the induction of T-cell exhaustion[30–33]. Exhausted T cells are characterized by an increased expression of inhibitory markers such as PD-1, TIM-3, TIGIT, CTLA-4, and LAG-3, and a progressive and hierarchical loss of function[30–33]. However, we found that Tph2 cells expressed high PD-1/*PDCD1* but low *HAVCR2* (TIM-3), *CTLA4*, and *LAG3*, and almost comparable levels of *TIGIT* with other Tph subsets, which suggests these cells are distinct from GPR56⁺CD4⁺ T cells and exhausted CD4⁺ T cells. Taken together, Tph2 cells appear to have a unique feature distinguishable from other "classical" Tph subsets. In future, it is necessary to elucidate the characterization of Tph2 cells infiltrated in local inflammatory sites.

It has been widely accepted that Tph cells promoting autoantibody production via IL-21 and play a pathogenic role in RA[7–12], SLE[13–17], or other autoimmune diseases[18–21]. However, there is only one study on Tph subsets in SLE[14]. Makiyama et al. have been demonstrated that Tph1 and Tph2 cells are expanded in the blood of active SLE patients and showed a positive correlation between CD38⁺ or HLA-DR⁺ Tph 1 cells and plasmablasts[14].

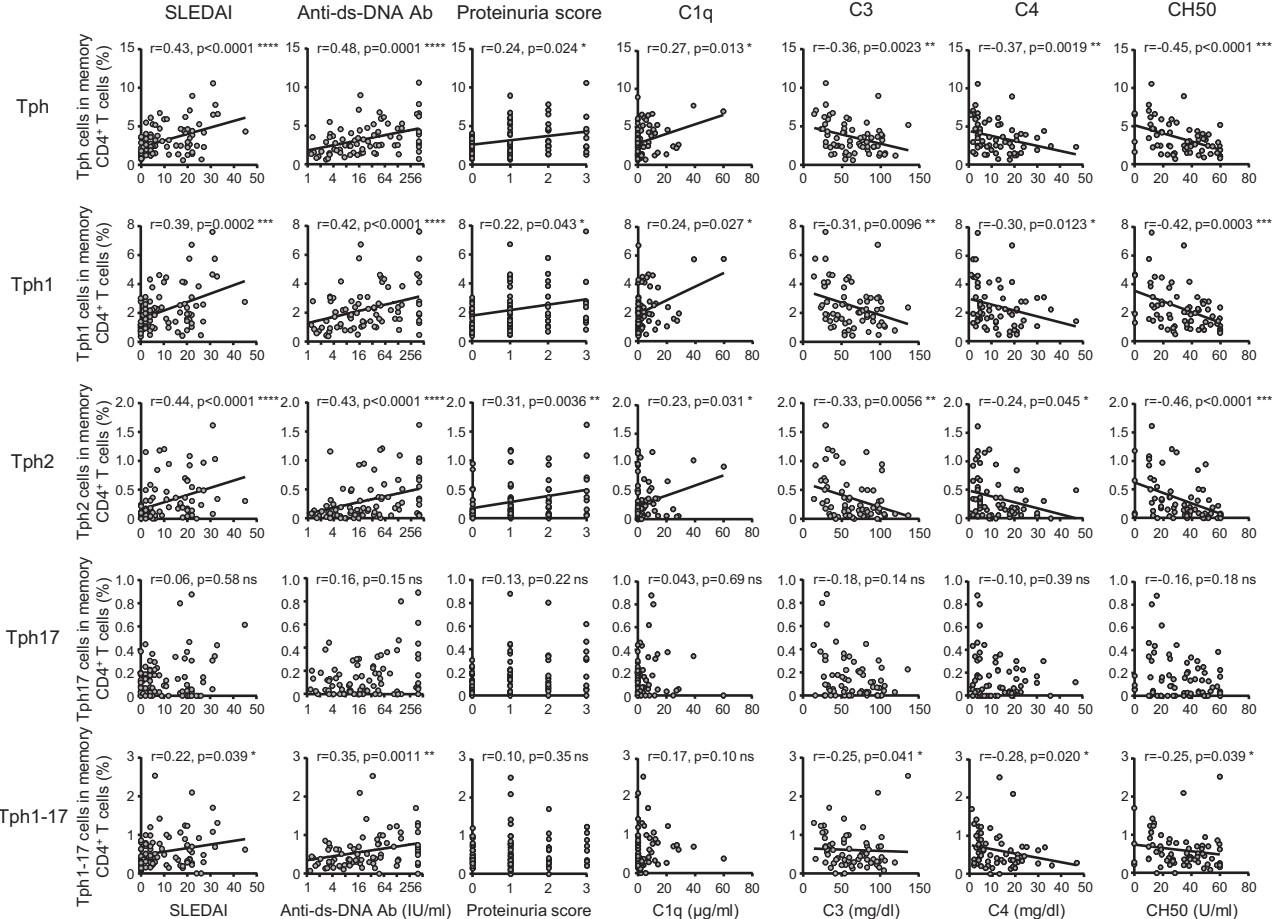

**Fig. 5 | The correlation between Tph subsets and disease activity in SLE patients.** The correlation between the frequency of each Tph subset within memory CD4$^+$ T cells and SLEDAI ($n = 85$), anti-ds-DNA antibody titers ($n = 84$), proteinuria scores ($n = 85$), C1q levels ($n = 82$), C3 levels ($n = 66$), C4 levels ($n = 65$), and CH$_{50}$ levels ($n = 68$) in total SLE patients. Spearman's rank correlation coefficients are shown. *$P < 0.05$, **$P < 0.01$, ***$P < 0.001$, ****$P < 0.0001$. ns: not significant.

Nevertheless, they did not show any relationship between Tph subsets and the disease activity of SLE. Therefore, until now, there is no report showing the involvement of Tph subsets in various clinical manifestations of SLE. In this study, we clearly demonstrated that both Tph1 and Tph2 subsets were markedly expanded in the blood of untreated new-onset SLE patients and that the frequency of Tph1 and Tph2 subsets were positively correlated with SLEDAI. Notably, the frequency of Tph1 subset was increased in patients with cutaneous and musculoskeletal manifestations. On the other hand, the frequency of Tph2 subset was increased in patients with lupus nephritis in addition to the above manifestations. Although Tph17 cells were shown to play a pathogenic role in psoriasis[34], Tph17 cells appear to play no or an only limited role in pathogenesis of SLE. Consequently, it is possible that Tph1 and Tph2 subsets are contributed to the complexity of clinical manifestations in SLE.

It has been reported that Tph cells express CCR2, CCR5, and CX3CR1 which play an important role in the cell migration into the inflammatory sites[7,8]. Nevertheless, there is no report on these chemokine receptor expressions in Tph subsets. We found in this study that the majority of Tph2 cells were CX3CR1-positives, while more than half of Tph1 cells expressed high CCR5 but low CX3CR1. The immunohistochemical data revealed that CXCL10 and CXCL11 were mainly expressed by basal keratinocytes while CXCL9 was detected in the dermal infiltrated macrophages[35] and that CXCR3$^+$ T cell were infiltrated into the inflammatory skin in SLE[36,37]. Additionally, skin tissues from SLE patients expressed higher levels of CCL2 and CCR2[36]. From these findings, it is highly probable that Tph1 cells preferentially migrate and infiltrate into skin lesions via CXCR3-CXCR3 ligands axis, CCR2-CCL2 axis, and CCR5-CCL3 axis in SLE.

By contrast, Tph2 cells appeared to be involved in lupus nephritis, cutaneous manifestation, and musculoskeletal manifestation via CX3CR1-CX3CL1 axis. The elevated levels of CX3CL1 expression and infiltration of CX3CR1-expressing CD16$^+$ monocytes have been found in glomeruli of SLE patients with proliferative lupus nephritis[38–41]. In addition, CD16$^+$ macrophages are the producers of CCL2 in kidneys of patients with lupus nephritis[42]. Based on these findings, it is plausible that CX3CR1-positive Tph2 cells can preferentially migrate into lupus kidney via CX3CR1-CXCL1 axis and contribute to the lupus nephritis in combination with CD16$^+$ monocytes.

In conclusion, we found that Tph1 subset with a B cell helper function is predominantly associated with cutaneous and musculoskeletal manifestations in SLE. In contrast, Tph2 subset is shown to possess low B cell helper activity but express cytotoxic function-related molecules and may be contribute to the tissue injury of lupus nephritis in addition to cutaneous and musculoskeletal manifestations in SLE. Our findings could provide new biomarkers for diagnosis and evaluation of disease activity and bring novel therapeutic approaches for each manifestation of SLE in future.

## Methods
### Patients
Eighty-five patients with SLE (median age 46 years (range 34–60), 88.2% female) and 33 HC (median age 40 years (range 29–50), 63% female) were prospectively included in this study. The new onset patients ($n = 27$) diagnosed as SLE within the past 6 months prior to enrollment had no treatment with major immunosuppressive therapies. In addition, untreated patients with RA ($n = 46$), p-SS ($n = 16$), AAV ($n = 27$), and IgG4-RD ($n = 37$) were

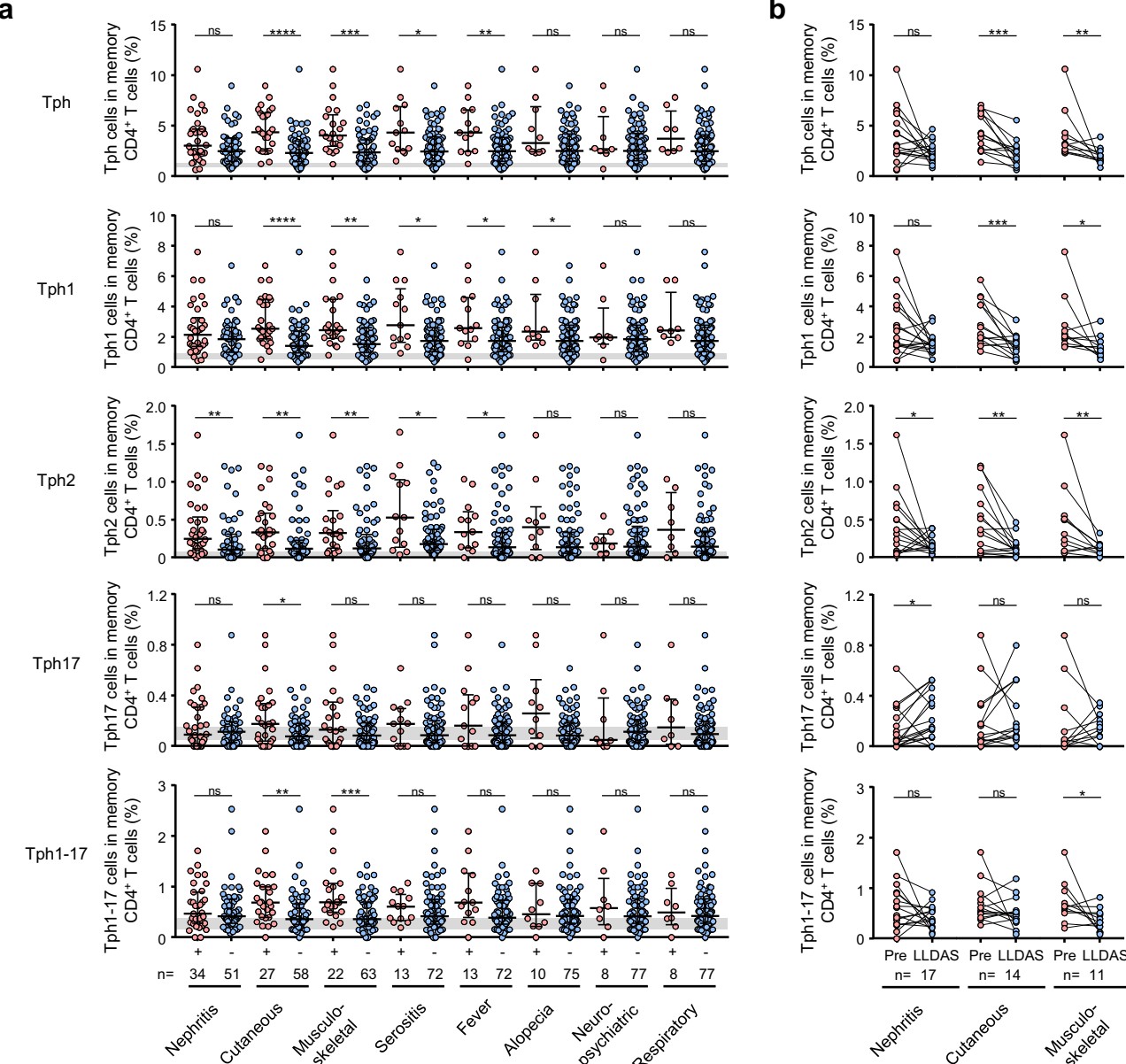

**Fig. 6 | The frequency of Tph subsets in the blood of SLE patients with various clinical manifestations. a** The frequency of each Tph subset in SLE patients (n = 85) with various manifestations including nephritis (n = 34), cutaneous (n = 27), musculoskeletal (n = 22), serositis (n = 13), fever (n = 13), alopecia (n = 10), neuropsychiatric (n = 8), and respiratory manifestations (n = 8). Data are the median ± IQR and the light gray area indicates the IQR of HC. Data were analyzed using Mann-Whitney U-test. *P < 0.05, **P < 0.01, ***P < 0.001, ****P < 0.0001, ns: not significant. **b** Proportions of each Tph subset before and LLDAS in SLE patients with nephritis (n = 17), cutaneous (n = 14), and musculoskeletal manifestation (n = 11). Data were analyzed using Wilcoxon's signed-rank test. *P < 0.05, **P < 0.01, ***P < 0.001, ****P < 0.0001, ns: not significant.

also included in this study for the comparison with untreated SLE patients. The pivotal clinical parameters and medications of patients with SLE or other autoimmune diseases are summarized in Supplementary Tables 3 and 4. We have confirmed that HC did not have autoimmune disease, severe allergic disorder, malignancy or infection. We reviewed the data of consecutive patients who visited Keio University Hospital from May 2015–December 2019, and were diagnosed with SLE, RA, p-SS, AAV, or IgG4-RD according to their respective classification criteria[43–47]. Clinical manifestations were defined according to the BILAG2004. Disease activity of SLE was evaluated using the SLE Disease Activity index (SLEDAI)-2K. For the validation cohorts, patients with autoimmune diseases and HC were enrolled at Keio University and written informed consent was obtained from all study participants. Ethical approval for this study was granted by the ethics committee of Keio University School of Medicine (protocol #20140335) and by

Mitsubishi Tanabe Pharma Corporation (protocol #H14018). All ethical regulations relevant to human research participants were followed.

## Flow cytometry and cell sorting

Heparinized blood samples were collected from patients with SLE, RA, p-SS, AAV, IgG4-RD, or HC. To determine Tph subsets, blood samples were stained with appropriate monoclonal antibodies (mAbs) and fixed by Phosflow Lyse/Fix Buffer (BD Biosciences, Franklin Lakes, NJ). FACS analysis was performed according to the methods described previously[48,49]. Flow cytometric analysis was conducted on BD LSRFortessa™ X-20 and analyzed by FlowJo ver.10 (TreeStar). Tph cells were defined as CXCR5⁻ memory CD4⁺ T cells (CD3⁺CD19⁻CD4⁺CD8⁻CD45RA⁻) expressed high levels of PD-1. Tph subsets were determined as follows: Tph1, CXCR3⁺CCR6⁻; Tph2, CXCR3⁻CCR6⁻; Tph17, CXCR3⁻CCR6⁺; Tph1-

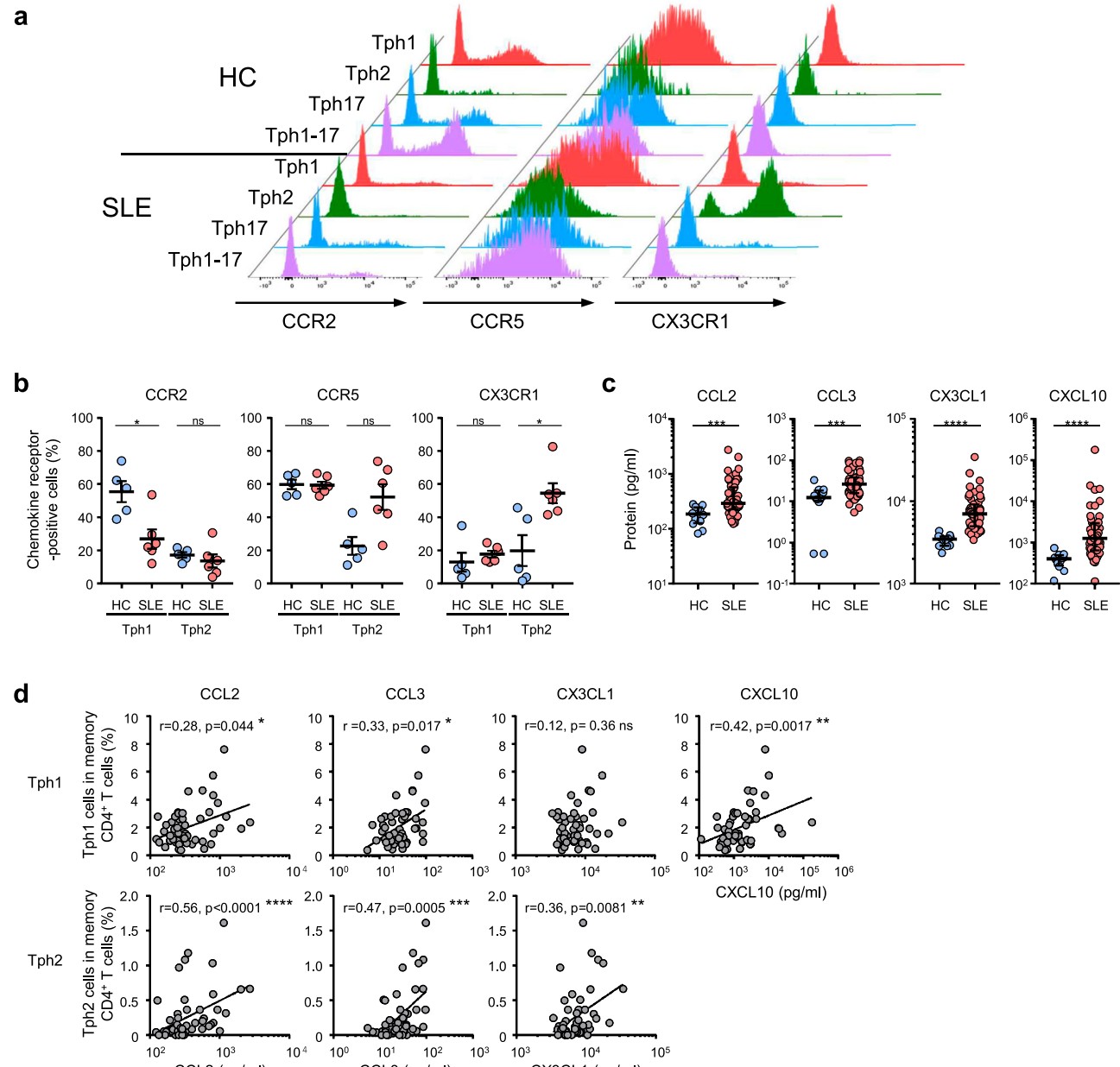

**Fig. 7 | Expression pattern of CCR2, CCR5, and CX3CR1 on each Tph subset. a** A typical expression pattern of the chemokine receptors on each Tph subset in HC and SLE patients. **b** The percentages of CCR2-, CCR5-, or CX3CR1-positive Tph1 or Tph2 subset in HC ($n = 5$) and SLE patients ($n = 6$). Data represent the mean ± SEM. Data were statistically analyzed using unpaired $t$-test. **c** The levels of CCL2, CCL3, CX3CL1, and CXCL10 in the serum of HC ($n = 12$) or SLE patients ($n = 51$). Data represent the median ± IQR. Data were statistically analyzed using Mann-Whitney $U$-test. **d** Correlation between the frequency of Tph subsets and the levels of CCL2, CCL3, CX3CL1, or CXCL10 in the serum of SLE patients ($n = 51$). Spearman's rank correlation coefficients are shown., *$P < 0.05$, **$P < 0.01$, ***$P < 0.001$, ****$P < 0.0001$. ns: not significant.

17, CXCR3⁺CCR6⁺ based on the expression of CXCR3 and CCR6[12]. In some experiments, cell surface expressions of CCR2, CCR5 and CX3CR1 or intracellular expressions of granzyme B and perforin were determined in each Tph subset. Intracellular staining was performed according to the method described previously[49]. FITC-conjugated anti-CD45RA (HI100), BV786-conjugated anti-CD3ε (SK7), PE-CF594-conjugated anti-CD3ε (UCHT1), PerCP-Cy5.5- or BUV395-conjugated anti-CD19 (HIB19), PE-Cy7- or BUV737-conjugated anti-CD4 (SK3), BUV563-conjugated anti-CD8 (RPA-T8), BV510-conjugated anti-PD-1 (EH12.1), PE-conjugated anti-CXCR3 (1C6/CXCR3), BV421-conjugated anti-CXCR5 (RF8B2), and PE/Cy7-conjugated anti-CCR5 (2D7) mAbs were obtained from BD Biosciences. BV711-conjugated anti-CXCR3 (G025H7), PerCP/Cy5.5-, APC-, or APC/Cy7-conjugated anti-CCR6 (G034E3), PE/

Dazzle594- or APC-conjugated anti-CCR2 (K036C2), PE-conjugated anti-CX3CR1 (2A9-1), APC-conjugated anti-perforin (dG9), and pacific blue-conjugated anti-granzyme B (GB11) mAbs were purchased from BioLegend (San Diego, CA). Cell sorting of Tph subsets (Tph1, Tph2, Tph17, and Tph1-17) was performed by Becton Dickinson FACS Aria II (BD biosciences). Naïve CD4⁺ T cells were isolated by using Naive CD4⁺ T cell Isolation Kit II (human) (Miltenyi Biotec, Bergisch Gladbach, Germany).

**Measurements of chemokines**

The concentrations of CCL2, CCL3, CCL4, CCL13, CCL20, CCL26, CXCL10, and CX3CL1 in the serum of SLE patients or HC were measured by electrochemiluminescence (ECL) assay, because of the higher sensitivity, lower level of detections, and wide dynamic range[17]. The ECL assay was

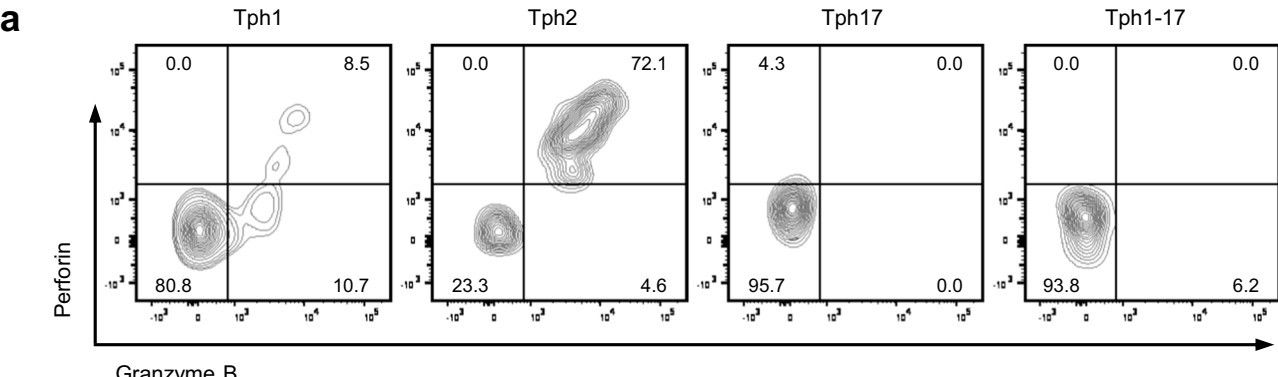

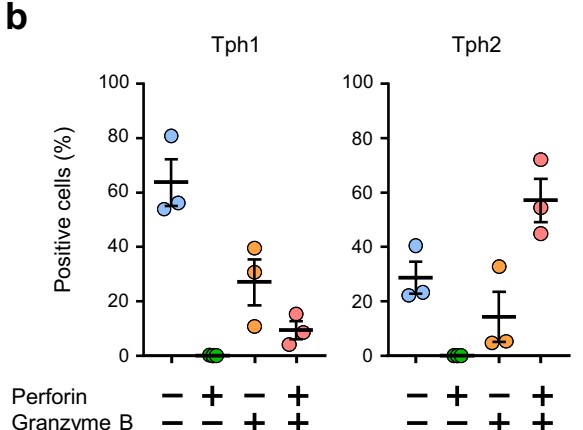

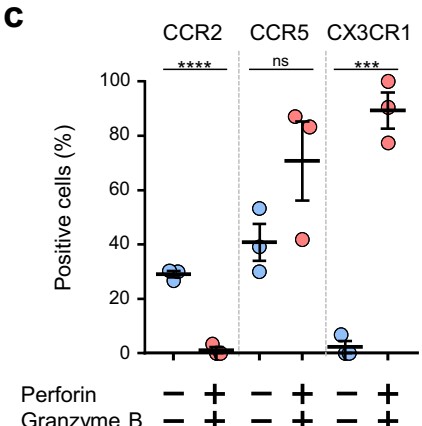

**Fig. 8 | CX3CR1-positive Tph2 cells express granzyme B and perforin. a** A typical expression pattern of granzyme B and perforin in each Tph subsets by intracellular staining. **b** The percentages of granzyme B$^+$ and/or perforin$^+$ cells in Tph1 and Tph2 subsets ($n$ = 3). Data represent the mean ± SEM. **c** The percentages of

granzyme B$^+$ and/or perforin$^+$ cells in CCR2-, CCR5- or CX3CR1-positive Tph2 cells ($n$ = 3). Data represent the mean ± SEM. Data were analyzed using unpaired $t$-test., \*\*\*$P$ < 0.001, \*\*\*\*$P$ < 0.0001. ns: not significant.

performed using Human Multiplex Ultra-Sensitive Kit and MESO Quick-Plex SQ 120 (Meso Scale Discovery, MSD, Gaithersburg, MA) and analyzed by Discovery Workbench 4.0.12 (MSD). The concentrations of CCL5, CXCL9, and CXCL11 in the plasma were measured by Cytokine Bead Array system and FCAP Array v3 (BD Biosciences). All experiments were performed according to the manufacturer's instruction with minimal modification and optimization.

### RNA sequencing and data processing
After cell sorting to TRIzol reagent (Thermo Fisher Scientific, Waltham, MA), total RNA was purified using RNA Clean and Concentrator (Zymo Research, Orange, CA). Standard quality control steps were included to determine total RNA quality using BioAnalyzer RNA pico kit (Agilent Technologies, Santa Clara, CA). Each purified total RNA obtained from each sample was subjected to a sequencing library construction using Smart-Seq Stranded Kit (Takara Bio, Shiga, Japan) according to the manufacture's protocol. The pooled libraries of the samples were sequenced using NovaSeq 6000 instrument (Illumine, San Diego, CA) with v1.5 reagents and a 150 bp paired-end configuration. To remove technical sequences, including adapters, polymerase chain reaction (PCR) primers, or fragments thereof, and quality of bases lower than 20, pass filter data of fastq format were processed by Cutadapt (V1.9.1, phred cutoff: 20, error rate: 0.1, adapter overlap: 1 bp, min. length: 75, proportion of N: 0.1) to be high quality clean data. The sequence reads were aligned to the human reference genome (GRCh38) using Hisat2 (v2.2.1). Subsequently, mapped reads were analyzed with the HTSeq (v0.6.1) software for estimating the expression of each gene. The expression values were converted to rank in ascending order. Heat maps show normalized gene expression values.

### Statistics and reproducibility
The results were shown as individual data with median ± interquartile range (IQR) or mean ± standard error mean (SEM). Statistical differences were calculated by Mann-Whitney $U$-test, Wilcoxon signed-rank test, or unpaired $t$-test as indicated. Correlations between two groups were analyzed using Spearman's rank correlation coefficient. Statistical significance was set at $P$ < 0.05. All statistical analyses including PCA were performed with GraphPad Prism software version 9.4.1 (GraphPad Prism Software Corporation, San Diego, CA).

### Reporting summary
Further information on research design is available in the Nature Portfolio Reporting Summary linked to this article.

### Data availability
RNA sequencing data have been deposited in the DNA Data Bank of Japan's BioProject (https://www.ddbj.nig.ac.jp/bioproject/index-e.html) and are accessible through the accession number PRJDB17562. The source data underlying the graphs in the article and supplementary information file can be found in Supplementary Data 1.

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

## Acknowledgements

We thank Yumi Ikeda for providing an excellent technical support. We also thank all the patients and healthy individuals who participated in this study.

## Author contributions

All authors listed on the manuscript have substantially contributed to this work. N.S. H.T. and S.T. designed research studies, conducted the experiments, and acquired the data. S.K. and F.M. analyzed the data of RNA sequencing. K.S. and K.Y. provided valuable suggestions for the study. J.K. S.S. Y.K. and M.A. conducted clinical evaluations. N.S. and K.C. analyzed the data, interpreted the results, and wrote the manuscript. K.C. and T.T. equally contributed to conceptualize and supervise the studies.

## Competing interests

The authors declare the following competing interests: Y.K. has received speaking or consultant fees from Asahikasei Pharma Corp. and Eli Lilly Japan KK. T.T. has received speaking or consultant fees from AbbVie GK, Bristol-Myers KK, and Chugai Pharmaceutical Co., Ltd. All other authors declare no competing interests.
