## [Peer Review File · Communications Biology]

Reviewers' comments:

Reviewer #1 (Remarks to the Author):

this is an interesting paper which presents information on subsets of TPH cells in pts with lupus. The data are clean and the manuscript has been presented clearly.

The manuscript is solely descriptive but there is limited someone can do with human samples. The subsets have been identified on the basis of two chemokine receptors and clinical correlations have been sought. Obviously, the ms would reach another league should functional data had been incorporated.

The discussion, although well written, is too long. I would like to see what do the authors think about the expanded TPH2 (cytotoxic) subset been expand in people without LN and shrunk in people with LN. Also, there should be some discussion to address the fact that cytotoxic responses are in general decreased in people with SLE.

Reviewer #2 (Remarks to the Author):

The study aimed to evaluate the T peripheral helper (TPH) cells that are thought to contribute to extra-follicular B cell activation and play a pathogenic role in autoimmune diseases. They found that TPH1 and TPH17 subsets expressed substantial levels of IL21, indicating B cell helper functions. Furthermore, they found that the TPH2 subset is a CD4+ cytotoxic T lymphocyte. In SLE patients, the frequency of TPH1 and TPH2 subsets is significantly increased and positively correlates with SLE disease activity. TPH1 cells are observed in cutaneous and musculoskeletal manifestations, while TPH2 cells are in lupus nephritis SLE patients.

The manuscript describes the differential correlation of these cells in the clinical manifestations of SLE. The study is well justified. It is novel and exciting since it proposes biomarkers that could be associated with the clinical manifestations of the disease.

The evidence strengthens the conclusions.

The statistical analysis is adequate.

However, this reviewer requests:

Include the following information in supplementary table 2

- a) the demographic, clinical and laboratory data of the groups of patients with RA, AAV, and IgG4-RD
- b) Laboratory variables: Hemoglobin, platelets, leucocytes, lymphocytes, monocytes, neutrophils, NLR, C1Q, C3, C4.
- c) Include the type of lupus nephritis of the patients
- d) Perform a correlation analysis of the subpopulations with the NLR.

Reviewer #3 (Remarks to the Author):

This is an interesting study to assess peripheral helper CD4 T cells which are rapidly becoming a cell of interest in autoimmune disease. In the setting of systemic lupus erythematosus, these authors provide more information about this subset and some potential indicators of active disease based on the populations of these cells.

Several comments to improve the paper:

1. It is not explained or clear why the authors chose to separate Tph populations further into subsets

based on CXCR3 and CCR6. Please provide more context to these chemokine receptors in the context of lupus and some explanation for why these were used to separate out Tph cells. Nor is it clear why they call the populations Tph1, Tph2, Tph1-17, and Tph17. The papers they reference suggest the CXCR3+ cells produce IFN γ which may mean there is some overlap with Th1 helper subtype and, thus, it makes some sense to call them Tph1, however this terminology is still somewhat confusing. Tph2 cells are higher in Batf which may mean they have some overlap with Th2 cells but do they produce IL-4 or have higher GATA3? If not, it might be better to use different terminology for the 4 different subsets as likewise, Tph17 do not seem to be similar to Th17 cells.

2. It would be interesting to do gene set enrichment analyses comparing these subsets to gene sets associated with Th1, Th2, Th17 to see if there is some overlap with these genes. If so, perhaps the question of terminology as discussed in #1 may make more sense.

3. It is important to recognize that PD-1 is associated with exhausted T cell phenotypes and typically seen in the setting of chronic antigen which would be present in a disease like lupus. Therefore, perhaps the high expression of PD-1 in the Tph2 subset is secondary to "exhaustion"/chronic antigen and these cells should actually not be thought of as the "classical" Tph since they do not produce IL-21 and are shown to have cytotoxic activity with high granzyme b and perforin 1.

4. While the percentage of cells is interesting and important to follow, it would be helpful to know whether the overall numbers change in the SLE patients, particularly with/without treatment (figure 4, figure 5, figure 6, supplemental figure 1).

Dear Reviewers,

We thank the editor and reviewers very much for reviewing our manuscript (COMMSBIO-23-3167T) entitled “Cytotoxic Tph subset with low B-cell helper functions and its involvement in systemic lupus erythematosus” and their detailed and constructive feedback. There are the point-by-point responses to the reviewers’ comments as follows.

Answers for the comments from Reviewer #1

Reviewer #1 (Remarks to the Author):

1. this is an interesting paper which presents information on subsets of TPH cells in pts with lupus. The data are clean and the manuscript has been presented clearly. The manuscript is solely descriptive but there is limited someone can do with human samples. The subsets have been identified on the basis of two chemokine receptors and clinical correlations have been sought. Obviously, the ms would reach another league should functional data had been incorporated.

We deeply appreciate the reviewer’s instructive suggestions.

2. The discussion, although well written, is too long.

Based on reviewer’s comments, we shortened “Discussion” for one page but maintained the original contents basically.

3. I would like to see what do the authors think about the expanded TPH2 (cytotoxic) subset been expand in people without LN and shrunk in people with LN.

As reviewer pointed out, there are patients with high frequency of Tph2 subset but no lupus nephritis (LN), or the LN patients with low Tph2 frequency. However, Tph2 frequency was significantly higher in not only LN patients but also patients with cutaneous or musculoskeletal manifestation. Therefore, the patients with high Tph2 subset but no LN are the patients with cutaneous or musculoskeletal manifestation. On the other hand, the cases of the LN patients with low Tph2 frequency are probably due to

reduction of circulating Tph2 subsets by migration and infiltration into kidney, the inflammatory sites. Thus, our results implied that Tph2 cells play a pathogenic role in LN. Since Tph2 cells may exhibit cytotoxicity in lupus kidney and play a pathogenic role in LN, we think that the frequency of Tph2 cells in the blood is one of bio-marker of LN. We also think that it would be important to analyze the time course changes of blood Tph subsets and their infiltration in lupus kidney. Thus, we discussed about the necessity of tissue analyses in future (p 16, line. 269-270).

4. Also, there should be some discussion to address the fact that cytotoxic responses are in general decreased in people with SLE.

It is known that SLE patients showed the reduced cytotoxicity of CD8 T cells which are called “exhaustion” (McKinney EF et al. *Nature* 523:612, 2015, and Lima G et al. *Clin Exp Immunol* 204:285, 2021). As Tph cells are expressing high levels of PD-1/*PDCD1*, this may mean a kind of “exhaustion”. Alternatively, as shown in following Additional Fig. 1, our RNA sequencing data revealed that Tph2 subset showed low expression of *HAVCR2* (TIM-3) and *CTLA4*, and almost comparable levels of *TIGIT* with other Tph subsets, suggesting that Tph2 cells are not exhausted (p 7, line 118-119). These results were newly added as Supplementary Fig. 1 in revised manuscript. Furthermore, we added the discussion about CD4⁺ T cell exhaustion (p 15, line 261- p16, line 269).

Additional Fig. 1 Gene expression of *HAVCR2*, *TIGIT*, and *CTLA4* in Tph subsets. Results are shown as the individual data and mean \pm SEM ($n=4$). The differences were analyzed by unpaired *t*-test (*: $P<0.05$, ns: not significant).

Answers for the comments from Reviewer #2

Reviewer #2 (Remarks to the Author):

The study aimed to evaluate the T peripheral helper (TPH) cells that are thought to contribute to extra-follicular B cell activation and play a pathogenic role in autoimmune diseases. They found that TPH1 and TPH17 subsets expressed substantial levels of IL21, indicating B cell helper functions. Furthermore, they found that the TPH2 subset is a CD4+ cytotoxic T lymphocyte. In SLE patients, the frequency of TPH1 and TPH2 subsets is significantly increased and positively correlates with SLE disease activity. TPH1 cells are observed in cutaneous and musculoskeletal manifestations, while TPH2 cells are in lupus nephritis SLE patients.

The manuscript describes the differential correlation of these cells in the clinical manifestations of SLE.

The study is well justified. It is novel and exciting since it proposes biomarkers that could be associated with the clinical manifestations of the disease.

The evidence strengthens the conclusions.

The statistical analysis is adequate.

We deeply appreciate the reviewer's instructive suggestions.

However, this reviewer requests:

1. Include the following information in supplementary table 2

According to the reviewer's opinions, we added following data in Supplementary Table 2 and made several new Supplementary Tables.

a) the demographic, clinical and laboratory data of the groups of patients with RA, AAV, and IgG4-RD

We added the demographic, clinical and laboratory data of the groups of patients with RA, p-SS, AAV, and IgG4-RD in Supplementary Table 4.

b) Laboratory variables: Hemoglobin, platelets, leucocytes, lymphocytes, monocytes, neutrophils, NLR, C1Q, C3, C4.

We added laboratory variables: hemoglobin, platelets, leucocytes (white blood cells), lymphocytes, monocytes, neutrophils, neutrophil lymphocyte ratio (NLR), C1q, C3, and C4 in Supplementary Table 3.

c) Include the type of lupus nephritis of the patients

We added the type of lupus nephritis of SLE patients in Supplementary Table 2.

d) Perform a correlation analysis of the subpopulations with the NLR.

We performed a correlation analysis between frequency of Tph subsets and the NLR by Spearman correlation coefficient. As shown in following figure, the frequency of whole Tph cells, Tph1 subset, and Tph2 subset were positively correlated with the NLR (Tph vs NLR $r=0.24$, $p=0.0277^*$, Tph1 vs NLR $r=0.24$, $p=0.0262^*$, Tph2 vs NLR $r=0.22$, $p=0.0420^*$, see following Additional Fig. 2). These results were shown as r values and p values in Supplementary Table 6.

Additional Fig. 2 The correlation between the frequency of each Tph subset and neutrophil lymphocyte ratio (NLR) in SLE patients ($n=85$). Spearman's rank correlation coefficients are shown ($*P<0.05$. ns: not significant).

Answers for the comments from Reviewer #3

Reviewer #3 (Remarks to the Author):

This is an interesting study to assess peripheral helper CD4 T cells which are rapidly becoming a cell of interest in autoimmune disease. In the setting of systemic lupus erythematosus, these authors provide more information about this subset and some potential indicators of active disease based on the populations of these cells.

We deeply appreciate the reviewer's instructive suggestions.

Several comments to improve the paper:

1. It is not explained or clear why the authors chose to separate Tph populations further into subsets based on CXCR3 and CCR6. Please provide more context to these chemokine receptors in the context of lupus and some explanation for why these were used to separate out Tph cells. Nor is it clear why they call the populations Tph1, Tph2, Tph1-17, and Tph17. The papers they reference suggest the CXCR3⁺ cells produce IFN γ which may mean there is some overlap with Th1 helper subtype and, thus, it makes some sense to call them Tph1, however this terminology is still somewhat confusing. Tph2 cells are higher in Batf which may mean they have some overlap with Th2 cells but do they produce IL-4 or have higher GATA3? If not, it might be better to use different terminology for the 4 different subsets as likewise, Tph17 do not seem to be similar to Th17 cells.

As it is well known, human Th cells were further separable into CXCR3⁺CCR6⁻Th1 cells, CXCR3⁻CCR6⁻Th2 cells, CXCR3⁻CCR6⁺Th17 cells, and CXCR3⁺CCR6⁺Th1-17 cells (Maecker HT et al. *Nat. Rev. Immunol.* 12:191, 2012). According to this classification, Tfh cells were shown to be separable into Tfh1, Tfh2, Tfh17, and Tfh1-17 subsets. Similarly, Tph cells were also separable into Tph1, Tph2, Tph17, and Tph1-17 subsets (Makiyama A, et al. *Rheumatology* 58:1861, 2019, Li J, et al. *Rheumatology* 58:2188, 2019, and Yamada H, et al. *Rheumatology* 60:451, 2021). Therefore, our study is not the first one to term Tph subsets based on CXCR3 and CCR6 expression. As similar to the reviewer's opinions, after the analyses in this study, we felt that Tph2 may be an unsuitable term because of high expression of granzyme B and perforin but low *IL21* and no detectable *IL4*. However, at least in this study, we had to use the term "Tph2" under the context of previous studies of Tph subsets. Please understand our situation.

- It would be interesting to do gene set enrichment analyses comparing these subsets to gene sets associated with Th1, Th2, Th17 to see if there is some overlap with these genes. If so, perhaps the question of terminology as discussed in #1 may make more sense.

The RNA sequencing analyses revealed that Th1 and Tph1 expressed *IFNG* and *CXCR3*. Th17 and Tph17 expressed *IL17A* and *CCR6*. Th1-17 and Tph1-17 expressed *IFNG*, *IL17A*, *CXCR3*, and *CCR6*. These Tph subsets expressed significant levels of *IL21* but Th subsets did not. These results suggest that the above mentioned Tph subsets (Tph1, Tph17, and Tph1-17) share key characteristics of respective Th subset. On the other hand, as noted in the comment No.1, Tph2 cells may be dissimilar to Th2 cells because of high expression of granzyme B/*GZMB* and perforin/*PRF1* but no detectable *IL4*. Additionally, the levels of *GATA3* in Tph2 cells were not so high as compared with those of other Tph subsets (see following Additional Fig. 3). We added the results of *GATA3* in Tph subsets in new Supplementary Fig.1

Additional Fig. 3 Gene expression of *GATA3* in Tph subsets.

Results are shown as the individual data and mean \pm SEM ($n=4$). The differences were analyzed by unpaired *t*-test (ns: not significant).

- It is important to recognize that PD-1 is associated with exhausted T cell phenotypes and typically seen in the setting of chronic antigen which would be present in a disease like lupus. Therefore, perhaps the high expression of PD-1 in the Tph2 subset is secondary to "exhaustion"/chronic antigen and these cells should actually not be thought of as the "classical" Tph since they do not produce IL-21 and are shown to have cytotoxic activity with high granzyme b and perforin 1.

Similar to the Reviewer's opinions, we also think that the high expression of PD-1 in the Tph2 subset may be "exhaustion". However, our RNA sequencing data revealed that Tph2 subset showed low expression of *HAVCR2* (TIM-3) and *CTLA4*, and almost comparable levels of *TIGIT* with other Tph subsets, suggesting that Tph2 cells are not exhausted (p 7, line 118-119). These results were newly added as Supplementary Fig. 1 in revised manuscript. Thus, we completely agree with Reviewer's comments that Tph2 cells may be distinguish from the "classical" Tph since they do not produce IL-21 but have cytotoxic activity with high granzyme B and perforin. We added these contents in Discussion (p 15, line 261- p16, line 269).

4. While the percentage of cells is interesting and important to follow, it would be helpful to know whether the overall numbers change in the SLE patients, particularly with/without treatment (figure 4, figure 5, figure 6, supplemental figure 1).

According to the Reviewer's comments, we added cell numbers of Tph subsets (cells/ μ l) in the blood of the SLE patients in Supplementary Tables 5, 7, 8, and 9 which are equivalent to Figs. 4, 5, 6, and Supplementary Fig. 1, respectively. On the other hand, the percentages of Tph cells or Tph subsets within memory CD4⁺ T cells were most frequently used to analyze their association with disease activities in previous studies (Makiyama A, et al. *Rheumatology* 58:1861, 2019, Li J, et al. *Rheumatology* 58:2188, 2019, and Caelli S, et al. *Nat Med* 25:75, 2019). One of the most important reasons for using percentages in memory CD4⁺ T cells would be the severe lymphopenia in SLE patients. Indeed, as shown in Supplementary Table 5, 7, 8, and 9, the absolute numbers of Tph subsets did not show clear changes when compared to the percentages in memory CD4⁺ T cells. This is probably due to lymphopenia in which the number of CD4⁺ T cells in the blood was markedly reduced in SLE patients.

REVIEWERS' COMMENTS:

Reviewer #1 (Remarks to the Author):

the authors have tried but only partially succeeded in addressed the raised comments.

Reviewer #2 (Remarks to the Author):

This reviewer thanks the authors for answering each and every question accurately.

I suggest acceptance of the manuscript.

Kind regards

Reviewer #3 (Remarks to the Author):

Thank you for incorporating the reviewer comments into your revision. Overall this paper provides some interesting insight into CD4 T cells within the disease of lupus.

Final thoughts:

1. This sentence lines 266- 268 in the manuscript: However, we found that Tph2 cells expressed high PD-1/PDCD1 but low HAVCR2 (TIM-3), CTLA4, and LAG3, and almost comparable levels of TIGIT with other Tph subsets, suggesting a different feature from GPR56+CD4+ T cells or exhaust CD4+ T cells.

Could be changed, for better readability to:

However, we found that Tph2 cells expressed high PD-1/PDCD1 but low HAVCR2 (TIM-3), CTLA4, and LAG3, and almost comparable levels of TIGIT with other Tph subsets, which suggests these cells are distinct from GPR56+CD4+ T cells and exhausted CD4+ T cells.

2. Also, just because the cells did not express high levels of Tim-3, CTLA4, LAG3, or TIGIT does not necessary mean the CD4 Tph2 cells not "exhausted" as T cells exhaustion is not homogeneous and often the T cells will have varying expression of inhibitory receptors on their surface as the review you cited by Miggelbrink et al discusses. In fact, terminally exhausted CD8 T cells, for example, are characterized by high levels of PD-1 on their surface and limited cytokine production and upregulation of granzyme B which is somewhat similar to the Tph2 CD4 T cells with the lack of IL-21 production and increased granzyme B expression. However, these nuances, while interesting to think about in the clinical relevance of these cells, are possibly outside the scope of this paper.

Dear Reviewers,

We heartily thank the editor and reviewers very much again for reviewing our manuscript (COMMSBIO-23-3167A) entitled “Cytotoxic Tph subset with low B-cell helper functions and its involvement in systemic lupus erythematosus” and their detailed and constructive feedback. There are the point-by-point responses to the reviewers’ comments as follows.

Answers for the comments from Reviewer #1

Reviewer #1 (Remarks to the Author):

The authors have tried but only partially succeeded in addressed the raised comments.

Thank you very much again for your previous valuable comments. On the basis of reviewer’s comments, we revied as possible as we can. However, within this study or within a short period, it is very hard to perform functional assays of Tph subsets or to determine Tph subsets in inflammatory sites. So, we discussed the necessity of these studies in future. Please understand our situation.

Answers for the comments from Reviewer #2

Reviewer #2 (Remarks to the Author):

This reviewer thanks the authors for answering each and every question accurately. I suggest acceptance of the manuscript. Kind regards

We deeply thank your reviewing for our manuscript.

Answers for the comments from Reviewer #3

Reviewer #3 (Remarks to the Author):

Thank you for incorporating the reviewer comments into your revision. Overall this paper provides some interesting insight into CD4 T cells within the disease of lupus.

We deeply appreciate the reviewer's valuable suggestions.

Final thoughts:

1. This sentence lines 266- 268 in the manuscript: However, we found that Tph2 cells expressed high PD-1/PDCD1 but low HAVCR2 (TIM-3), CTLA4, and LAG3, and almost comparable levels of TIGIT with other Tph subsets, suggesting a different feature from GPR56+CD4+ T cells or exhaust CD4+ T cells.

Could be changed, for better readability to:

However, we found that Tph2 cells expressed high PD-1/PDCD1 but low HAVCR2 (TIM-3), CTLA4, and LAG3, and almost comparable levels of TIGIT with other Tph subsets, which suggests these cells are distinct from GPR56+CD4+ T cells and exhausted CD4+ T cells.

We agreed with the Reviewer's opinion and revised as "which suggests these cells are distinct from GPR56+CD4+ T cells and exhausted CD4+ T cells"

2. Also, just because the cells the cells did not express high levels of Tim-3, CTLA4, LAG3, or TIGIT does not necessary mean the CD4 Tph2 cells not "exhausted" as T cells exhaustion is not homogeneous and often the T cells will have varying expression of inhibitory receptors on their surface as the review you cited by Miggelbrink et al discusses. In fact, terminally exhausted CD8 T cells, for example, are characterized by high levels of PD-1 on their surface and limited cytokine production and upregulation of granzyme B which is somewhat similar to the Tph2 CD4 T cells with the lack of IL-21 production and increased granzyme B expression. However, these nuances, while interesting to think about in the clinical relevance of these cells, are possibly outside the scope of this paper.

We explained the typical exhausted T cells expressed high levels of inhibitory molecules and therefore Tph2 cells appear to be distinguishable from the typical exhausted cells. On the other hand, similarly with the Reviewer's opinion, we also think that Tph2 cells may be one of exhausted-typed cells because they express higher levels of PD-1 and cytotoxic molecules but showed low producers of IL-21. However, in this manuscript, we would like to mention predominantly that there is a unique cytotoxic Tph subset distinct from IL-21-producing Tph cells. Thank you very much for your valuable suggestion.